# Mouse brain transcriptome responses to inhaled nanoparticulate matter differed by sex and *APOE* in *Nrf2-Nfkb* interactions

Amin Haghani[1], Mafalda Cacciottolo[1], Kevin R Doty[2], Carla D'Agostino[1], Max Thorwald[1], Nikoo Safi[1], Morgan E Levine[3], Constantinos Sioutas[4], Terrence C Town[2], Henry Jay Forman[1], Hongqiao Zhang[1], Todd E Morgan[1], Caleb E Finch[1,5]*

[1]Leonard Davis School of Gerontology, University of Southern California, Los Angeles, United States; [2]Zilkha Neurogenetic Institute, Department of Physiology and Neuroscience, Keck School of Medicine of the University of Southern California, Los Angeles, United States; [3]Department of Pathology, Yale School of Medicine, New Haven, United States; [4]Department of Civil and Environmental Engineering, Viterbi School of Engineering, University of Southern California, Los Angeles, United States; [5]Dornsife College, University of Southern California, Los Angeles, United States

*For correspondence:
cefinch@usc.edu

Competing interests: The authors declare that no competing interests exist.

**Abstract** The neurotoxicity of air pollution is undefined for sex and *APOE* alleles. These major risk factors of Alzheimer's disease (AD) were examined in mice given chronic exposure to nPM, a nano-sized subfraction of urban air pollution. In the cerebral cortex, female mice had two-fold more genes responding to nPM than males. Transcriptomic responses to nPM had sex-*APOE* interactions in AD-relevant pathways. Only *APOE*3 mice responded to nPM in genes related to Abeta deposition and clearance (*Vav2, Vav3, S1009a*). Other responding genes included axonal guidance, inflammation (AMPK, NFKB, APK/JNK signaling), and antioxidant signaling (NRF2, HIF1A). Genes downstream of NFKB and NRF2 responded in opposite directions to nPM. *Nrf2* knockdown in microglia augmented NFKB responses to nPM, suggesting a critical role of NRF2 in air pollution neurotoxicity. These findings give a rationale for epidemiologic studies of air pollution to consider sex interactions with *APOE* alleles and other AD-risk genes.

## Introduction

Air pollution is a major global environmental risk factor of morbidity and mortality across the human lifespan (*Landrigan et al., 2018*; *Shaffer et al., 2019*; *Finch, 2018*). Air pollution exposure is also associated with neurodegeneration, accelerated cognitive decline of aging and increased risk of Alzheimer's disease (AD) (*Kulick et al., 2020*; *Calderón-Garcidueñas et al., 2020*). However, little is known of interaction of air pollution neurotoxicity for sex and *APOE* alleles and other AD risk factors (*Finch and Kulminski, 2019*).

Epidemiological studies of air pollution neurotoxicity have not identified interactions of gender by *APOE* alleles. Findings are typically 'adjusted or controlled' for gender differences (*Clifford et al., 2016*; *Chen and Schwartz, 2009*; *Ailshire and Clarke, 2015*; *Gatto et al., 2014*). In the WHIMS cohort of elderly women, *APOE*4 homozygotes had a greater risk of dementia and accelerated cognitive decline (*Cacciottolo et al., 2017*). The *APOE*4 vulnerability for accelerated cognitive aging was recently extended to ozone, as well as PM10 and PM2.5 in a large sample of both

sexes from New York City (*Kulick et al., 2020*). A recent study from China suggested greater male vulnerability to air pollution for verbal test deficits (*Zhang et al., 2018*). Sex-*APOE* interactions for air pollution neurotoxicity remain undefined. In a small sample from polluted Mexico City, *APOE4* heterozygous females with high BMI had higher risk of severe cognitive deficit than other groups (*Calderón-Garcidueñas and de la Monte, 2017*). Developmental air pollution exposure has received greater attention for gender because of the consistent male excess vulnerability for behavioral and cognitive deficits in the pre-adolescent and young adult (*Chiu et al., 2013*; *Sunyer et al., 2015*).

Mouse models have not addressed sex and *APOE* in responses to air pollution. Our initial study examined female EFAD (*APOE*-TR/5xFAD-Tg$^{+/-}$) mice carrying transgenes for familial AD genes (5xFAD) together with human *APOE* alleles by targeted replacement (*APOE*-TR), which had *APOE*-e3$^{+/+}$ (E3; *APOE3*) or *APOE*-e4$^{+/+}$ (E4; *APOE4*) genotype. Consistent with WHIMS findings, E4FAD female mice accumulated more brain amyloid in response to nPM than the E3FAD (*Cacciottolo et al., 2017*). However, for ozone exposure, male *APOE*-TR showed the converse of greater behavioral impairments in *APOE3* than *APOE4* (*Jiang et al., 2019*). For further study of both sexes, we examined the cerebral cortex transcriptomic responses of *APOE*-TR and wildtype mice (C57BL/6J) by RNAseq for the main regulators of air pollution toxicity in AD pathways.

We focused on genomic pathways mediated by NRF2 and NFKB, which responded to air pollution in our prior studies (*Zhang et al., 2012*; *Woodward et al., 2017a*). These redox-sensitive transcription factors control hundreds of genes that mediate cellular responses to oxidative stress and immunity. They respond to oxidative stress, tobacco smoke, traumatic brain injury, and ischemic stroke, and are altered by aging and AD (*Sivandzade et al., 2019*). NRF2 downstream genes include antioxidants (e.g. glutathione, thioredoxin), anti-inflammatory cytokines (e.g. IL10), phase I and II xenobiotic detoxifying enzymes (e.g. CYP450) and free radical scavengers (*Sandberg et al., 2014*). The NFKB family transcriptionally regulates the expression of immune related proteins including cytokines (e.g. TNFA, IL1A, IL1B), antigen presentation proteins (e.g. MHCI, β2-microglobulin), chemokines (e.g. MCP1, MIP1), adhesion molecules (e.g. ICAM1, VCAM1), inducible nitric oxide synthase (INOS), and proapoptotic (e.g. BIM, BAX) or antiapoptotic proteins (e.g. XIAP, BCL2) (*Hayden and Ghosh, 2012*). The complex interplay of NRF2 and NFKB signaling pathways can alter the balance of anti-oxidative or inflammatory responses, depending on the type of stress, and target cell or tissue (*Sivandzade et al., 2019*).

Sex and *APOE* alleles can also alter NRF2 and NFKB activities, as shown for the larger response of female mice for hepatic NRF2 activation by phenobarbital and oxazepam and other xenotoxins (*Rooney et al., 2018a*). NRF2 downstream genes including *Gsta2*, *Ho1*, and *Nqo1* showed lower hepatic expression in *APOE*4-TR than *APOE*3-TR mice (*Graeser et al., 2011*). We therefore examined both sex and *APOE* allele for interactions with NRF2/NFKB responses of air pollution neurotoxicity.

## Results

To define brain transcriptional responses of air pollution and interactions with sex and *APOE* alleles, we examined responses of adult C57BL/6J (wild type, 'B6') and B6 mice carrying human *APOE3* and *APOE4* alleles by targeted replacement (*APOE*-TR) to nPM, a subfraction of ultrafine PM (PM0.2). Three independent exposures used different batches of nPM at specified durations of exposure (details on sample collection and chemical composition in *Figure 1—figure supplement 1* and *Zhang et al., 2019*). In vitro studies on BV2 microglia examined the role of NRF2 and NFKB in responding inflammatory pathways.

### Cerebral cortex transcriptome responses to nPM

Differentially expressed genes (DEGs) were analyzed by RNAseq for nPM responses. Stratification by *APOE* and sex was done subsequently to establish general effects. The multivariate model of combined B6 and *APOE*-TR data was adjusted statistically for sex, *APOE* genotype, and different nPM batches of the two exposures. For p=0.005, there were 140 DEGs (118 increased, 22 decreased) responses to nPM (*Figure 1A*). Ingenuity pathway analysis (IPA) of responding pathways included synapse function (e.g. axonal guidance, calcium signaling, endocannabinoid neuronal synapse), inflammation (e.g. AMPK, SAPK/JNK), circadian rhythm, NRF2 mediated antioxidant response, and

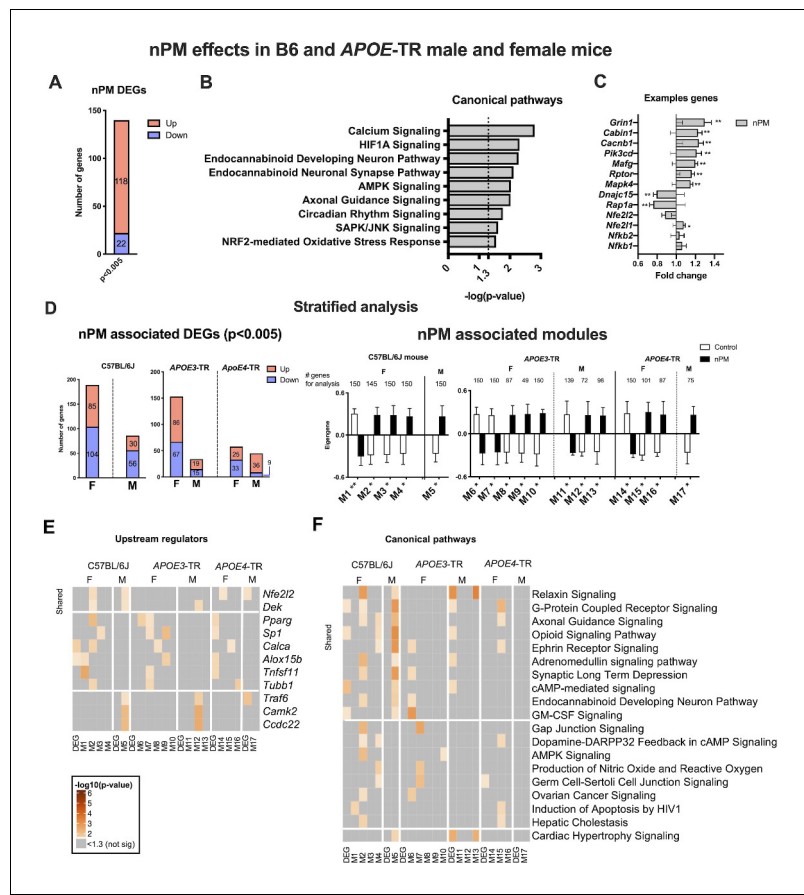

**Figure 1.** Cerebral cortex transcriptome responses to nPM in B6 and *APOE*-TR mice. (**A**) Multivariate differential expression analysis of nPM responses in combined data from the independent exposures of C57BL/6J (**B6**) and *APOE*-TR. Covariates included sex, *APOE* genotype, and nPM. DEGs identified at p-value, 0.005. (**B**) Canonical pathways associated with nPM DEGs. (**C**) Examples of nPM associated DEGs. (**D**) Sex- and *APOE*-stratified DE and WGCNA modules associated with nPM responses. Male, M; Female, F. The top 150 genes of modules (kME inter-module connectivity) were used for IPA analysis. Significance was calculated from the Pearson correlation of eigengene of the modules with nPM. (**E**) Upstream regulators and F) canonical pathways associated with nPM transcriptome responses in B6 and *APOE*-TR mice. Solid horizontal lines separate responses that are shared and sex-specific. Heatmaps were sorted by the sum of -log10 (p-values) in each row. p-values<$10^{-6}$ were converted to $10^{-6}$ for better visualization; grey, not significant. RNAseq sample size was 4/genotype/sex/treatment.

The online version of this article includes the following figure supplement(s) for figure 1:

**Figure supplement 1.** Chemical characterization (A) and cellular activity (B) of nPM batches collected after 2017 in this study.

**Figure supplement 2.** Top canonical pathways and potential upstream regulators of nPM associated genes in C57BL/6J mouse.

**Figure supplement 3.** nPM associated changes in male and female *APOE*-TR mouse.

**Figure supplement 4.** qPCR validation of selected genes from RNAseq, with GAPDH as reference gene, showing similar direction and scale of response to nPM by qPCR and.

hypoxia-inducible factor 1-α (HIF1A) signaling (*Figure 1B*). The top DEGs include *Grin1* (+20%) and *Rap1a* (−20%) (*Figure 1C*).

RNAseq data were stratified to identify sex- and *APOE*-specific nPM responses by linear models and by weighted gene co-expression (WGCNA). Females had more DEG than males for both B6 and *APOE*-TR mice, by up to two-fold (*Figure 1D*). Females of *APOE*3 and B6 had the most nPM responding genes (153 vs 189, respectively). Gene modules identified by WGCNA also had more female responses to nPM for B6 and *APOE*-TR (*Figure 1D*). Modules were constrained to a

maximum of 150 hub genes, based on connectivity (Materials and methods). Both analyses (DE, WGCNA) showed more nPM-responding gene responses for *APOE*3 than *APOE*4.

Upstream regulators and canonical pathways were identified by IPA for sex-specific and shared nPM responses. The top upstream candidate was *Nfe2l2* (*Nrf2*), a regulator of Phase II detoxification (*Figure 1E*, *Figure 1—figure supplements 2–3*), which had the strongest associations for B6 and *APOE*4. Sexes differed in immune-related upstream regulators of gene responses. Female-specific responses included *Pparg* (peroxisome proliferator activated receptor gamma), *Sp1* (specificity protein1 transcription factor), and *Tnfsf11* (TNF superfamily 11). Male-specific responses included *Traf6* (TNF associated receptor factor 6), *Camk2* (regulator of synaptic plasticity and AMPA glutamate receptors), and *Ccdc22* (regulator of NFKB signaling by interaction with COMMD proteins). These results paralleled the enrichment of NRF2 and immune response pathways in the combined multivariate model above.

Stratified analysis by *APOE* and sex for canonical pathways showed nPM responses of neuronal pathways; for example G-protein-coupled receptors, axonal guidance, ephrin receptors, synaptic long-term depression, and endocannabinoid development neuron pathway (*Figure 1F*). Other nPM responding genes were related to relaxin, GM-CSF, and c-AMP-mediated signaling. Female-specific responses include genes associated with the following pathways: AMPK, dopamine-PARPP32 feedback in cAMP, gap junction signaling, and nitric oxide production. Male-specific responses in both mouse strains were enriched for 'cardiac hypertrophy' signaling; for example, *Elk1* (transcription factor) and *Hsp27* (*Figure 1F*). *APOE*3 and *APOE*4 of both sexes had different inflammatory responses for NFKB, IL6, CREB, and IL22 pathways (*Figure 1—figure supplement 3*). Cell-type deconvolution analysis of RNAseq also showed *APOE* and sex-specificity for microglial and astrocyte responses to nPM (*Figure 1—figure supplement 3B*).

## Baseline effects of *APOE4* allele and the overlap with nPM responses

Baseline differences by *APOE* in non-exposed controls were analyzed by sex in two steps. The combined multivariate model showed 133 DEGs differed in baseline *APOE* allele effect (5% FDR) (*Figure 2A*). These DEGs were enriched for immune-related pathways including rheumatoid arthritis, granulocyte adhesion, IL10, and NFKB signaling. *APOE*4 baseline differences included pathways of glutamate metabolism, and production of nitric oxide, superoxide and hydrogen peroxide, the LXL/RXR pathway of cholesterol efflux, and atherosclerosis.

In stratified analysis, males had 60% more DEGs differing by *APOE* alleles (male, 75 genes; female, 45 genes) (*Figure 2B*, *Figure 2—figure supplement 1*). For WCGNA modules differing by *APOE* alleles, males had 3-fold excess (male 12 modules; female, four modules). Subsets of DEG (females, 70 DEGs; males, 37) had 7% overlap with nPM responding genes (*Figure 2C–D*), suggesting convergent effects of *APOE*4 allele and nPM. For females, the shared responding genes involved metabolic pathways (glycolysis, oxidative phosphorylation) and DNA repair (HMGN1). The male overlap involved a different set of genes related to iron homeostasis, telomere extension, and immune response (TREM1, IL11, JAK signaling).

AD-related pathways differed by sex and *APOE* alleles for nPM responses (*Figure 3A–B*). Only female *APOE*3 responded to nPM in five AD pathway genes for amyloid precursor protein (APP) processing and for tau: *App*, *Bace1*, *Psen1* (*Figure 3A*); *Tau* and *Gsk3b* (*Figure 3B*). In contrast to mRNA changes in amyloidogenesis genes, the levels of amyloid peptides (Aβ40 and Aβ42) were not affected by nPM exposure in the cerebral cortex of APOE-TR mice (*Figure 3C*). However, Aβ40 peptide had a 50% lower baseline in males than females of both *APOE*3 and *APOE*4-TR mice (*Figure 3C*). Aβ40 peptide also had a significant negative correlation with mRNA levels of amyloidogenesis pathway including *App* (r, −0.65) and *Psen1* (r, −0.51 to −0.67) for both *APOE*3 and *APOE*4-TR cerebral cortex (*Figure 3D*).

About 10% of genes related to amyloid clearance (5/46) differed by *APOE* or nPM. For amyloid clearance genes, only *APOE*3 carriers in both sexes responded to nPM (*Figure 3E*). Three genes responded to nPM with sex differences only in *APOE*3. For *APOE*3 females, *Vav2* (+50%); *Vav3* (decreased −50%); the Vav guanine nucleotide exchange factors mediate phagocytosis of fibrillar Aβ (*Wilkinson et al., 2006*). For *APOE*3 males, *S100a9* (−60%), also known as *Mrp14*, regulates microglial phagocytosis of fibrillar Aβ (*Kummer et al., 2012*). Baseline expression of two complement genes was higher in *APOE*3 than *APOE*4: *C3* (10-fold), *C3ar1* (+50%).

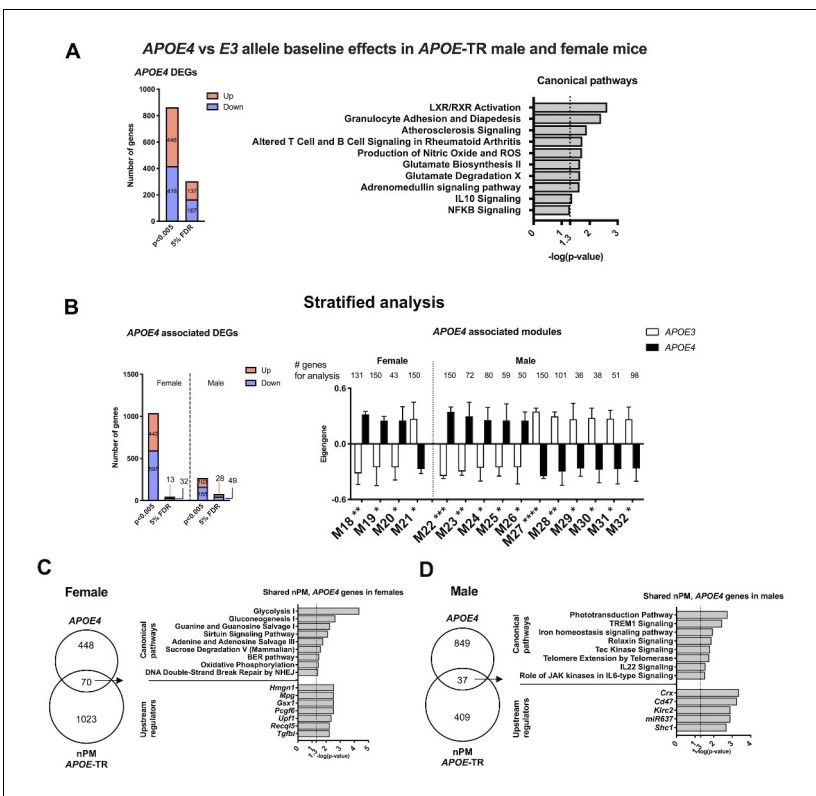

**Figure 2.** *APOE* allele baseline differences of RNA in cerebral cortex. (A) Differential expression analysis of *APOE*4- vs *APOE*3-TR, at 5% FDR and p-value, 0.005. (B) WGCNA modules associated with *APOE*4 allele. IPA of the top 150 genes of the modules identified by kME (inter-module connectivity). Significance was calculated from the Pearson correlation of eigengenes for modules with *APOE*4 allele. *p<0.05; **p<0.01; ***p<0.001; ****p<0.0001. IPA analysis of overlapped genes between baseline differences by *APOE* allele and nPM response in females (C) and males (D). The genes in each group are a combination of identified genes based on DE and WGCNA. RNAseq sample size was 4/genotype/sex/treatment. Detailed IPA analysis of *APOE* allele baseline DEGs (*Figure 2—figure supplement 1*).

The online version of this article includes the following figure supplement(s) for figure 2:

**Figure supplement 1.** Upstream regulators and canonical pathways associated with *APOE4* baseline difference in male and female *APOE*-TR mouse.

## Sex- and *APOE*- specific nPM-mediated NFKB responses

Next, we examined genes of the NFKB pathway, which regulate pro-inflammatory responses to nPM, as shown for responses of wildtype mice (B6 males) in hippocampus to nPM (*Woodward et al., 2017a*). In cerebral cortex, *APOE*-TR mice responded to nPM with a subset of genes downstream of NFKB (13%, 8/133) that differed by sex and *APOE* (*Figure 4A*). Two clusters of nPM responses were identified by Principal Component Analysis for these 133 NFKB downstream genes: Principal component (PC) 2 (20% of variance, nPM: sex interaction) and PC4 (2.5% of variance nPM:*APOE4* interaction) (*Figure 4B*).

Cytokine protein levels were examined for association with these PCs in cerebral cortex. PC2, but not PC4, was correlated with TNFA (r = −0.39, p=0.02), IL1B (r = 0.51, p=0.002), and CXCL1 (r = 0.43, p=0.01) proteins (*Figure 4C*). Only females responded to nPM for these cytokines. These RNA and protein responses are notable for consistent sex-specific inflammatory responses to nPM.

## Sex- and *APOE* allele-specific NRF2 responses

NRF2 downstream responses to nPM differed by sex and *APOE* (*Figure 5A*). The 60 responding genes included *Gpx3*, and *Gstp1*, *Jun*, *Nfe2l1* (*Nrf1*), and several *Maf* family transcription factors. A subset of gene responses was validated by qPCR, for example *Nrf1* (*Figure 5B*; *Figure 1—figure*

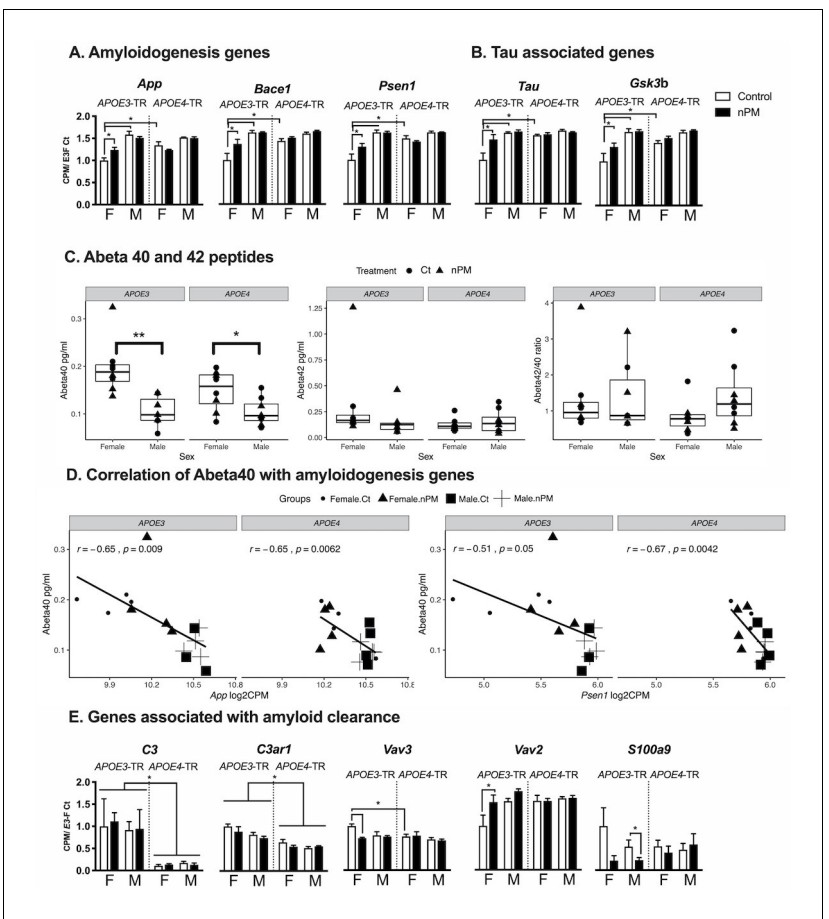

**Figure 3.** AD-associated gene responses to nPM in cerebral cortex. (**A**) Amyloidogenic pathway responses. Female *APOE3* had largest nPM response. (**B**) *Tau* and its kinase, *Gsk3b*. (**C**) Levels of Aβ40 and Aβ42 peptides did not respond to nPM. However, females had higher Aβ40 (pg/ml) of both *APOE3* and *APOE4* mice. \*\*p<0.01, \*p<0.05 in t-test. (**D**) Aβ40 peptide was negatively correlated with amyloidogenesis gene expression including App (r = −0.65), and Psen1 (r = −0.51 to −0.67). The expression was reported as Log2 count per million (cpm). (**E**) Aβ-amyloid clearance pathway responses to nPM. A small subset (10%, 5/46) of amyloid clearance genes differed by *APOE* or nPM (genes identified in the IPA database for phagocytosis, proteolysis, degradation, deposition). Only *APOE*3 responded to nPM. Mean ± SEM. ANOVA; FDR multiple test correction. \* Adj. p-value, 0.05. Sample size: 4/genotype/sex/treatment.

supplement 4). Female B6 and *APOE*-TR had 2-fold or more *Nrf2* downstream genes responding to nPM. PC2 is associated with nPM for interactions with sex (p=0.01) and *APOE* (p=0.02), 6.4% of the variance, mainly associated with *APOE3* females (*Figure 5C*). The strong inverse correlation of *Nrf2* PC2 with *Nfkb* PC2 (*Figure 5E*, r = - 0.95, p=0.0001) suggests crosstalk between these transcriptional factors during exposure to nPM.

## Inhibitory effects of NRF2 on NFKB response to nPM

The relationship of NRF2 and NFKB responses of nPM was further explored in an independent dose-response experiment. The duration of inhalation exposure was only 3 weeks to assess the initial responses of nPM by male C57BL/6 mice. After 3-week exposure to 300 µg/m$^3$ nPM, the cerebral cortex had nuclear translocation of NRF2 protein (+50%) and increased cytosolic NFKBP65 (+25%) (*Figure 6A*). Downstream of NRF2, the rate limiting enzyme of glutathione synthetase (GCLC) had dose-dependent increase correlated with *Nrf2* mRNA (r, 0.6, p=0.005) (*Figure 6B*). *Nrf2* and *Nfkb* responses of B6 male mice at three nPM doses had opposing changes of increased *Nrf2* mRNA, but decreased *Nfkbp65*, *Nfkbp50* mRNA and IL2 protein, all with dose-dependence (*Figure 6C*).

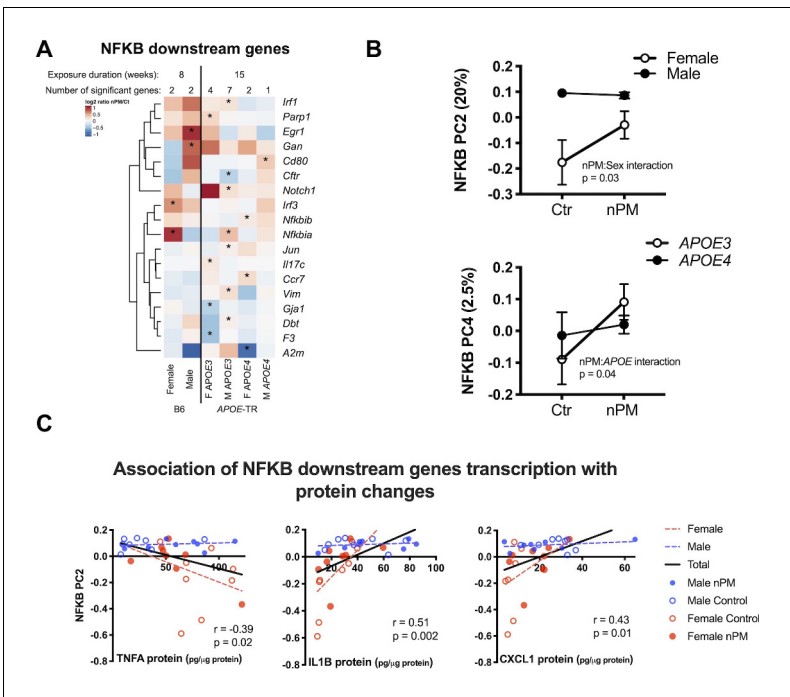

**Figure 4.** nPM induced inflammatory responses with sex- and *APOE* specificity. (**A**) Stratified analysis of NFKB downstream genes responses to nPM. The combined IPA datasets included 133 NFKB downstream genes. (**B**) Principal component analysis of 133 NFKB downstream genes in *APOE*-TR: PC2 (20% of total variance) was associated with nPM: sex interaction; PC4 (2.5% of variance), associated with nPM:*APOE* interaction. (**C**) Protein levels of genes downstream of NFKB were correlated with PC2: positive correlations for CXCL1 and IL1B; inverse correlation with TNFA. Only females responded to nPM. Sample size of 4/genotype/sex/treatment.

Nrf2 and Nfkb interactions were examined in BV2 microglia in vitro during acute (6 hr) exposure to nPM. Partial knockdown of *Nrf2* (−40%) by siRNA increased the *Nfkb* mediated responses of nPM (*Figure 6D*), with 30% higher *Nfkbp50* mRNA, and 200% higher mRNA of *Inos*, *Il1b* and *Il6*.

## NRF2 and NFKB are potential regulators of the nPM responses in the cerebral cortex and mixed glial culture

The top canonical pathways related to nPM exposure in the cerebral cortex of adult mice comprised of calcium signaling, HIF1α signaling, circadian rhythm pathway, AMPK signaling, SAPK/JNK signaling, endocannabinoid pathways and NRF2 oxidative stress responses (*Figure 1A*). These highly interconnected pathways comprise a larger network of oxidative and inflammatory responses (*Figure 7*). Thus, the hub regulators of nPM responses could broadly affect these pathways. Using IPA analysis, we built two networks from the NRF2 and NFKB downstream genes in the identified canonical pathways. Exposure to nPM caused expression changes in four of these genes in the cerebral cortex of adult mice: *Smarca4* (+25%), *Cftr* (+25%), *Hdac1* (−65%), and *Vegfc* (−65%).

We hypothesized that these networks are among the initial responses to nPM exposure and contributed to chronic damage at later stages. The transcriptional responses of these networks were examined in mixed glial culture after 24 hr exposure to 10 µg/ml nPM (Dataset from prior nPM studies *Woodward et al., 2017a*). Major increases of both *Nrf2* (300%) and *Nfkb* (400%) mRNA were induced by acute nPM exposure (*Figure 7*). A large portion of genes from the constructed networks was also upregulated by nPM: 17 NRF2 and 30 NFKB DEGs from these downstream networks. These genes were related to all selected signaling pathways: HIF1A, calcium, AMPK, SAPK/JNK, and the endocannabinoid system. Several of these genes are also transcriptional regulators that could initiate further downstream changes, body-wide. Some of these genes include *Brca1* (+30%), *Atf4* (350%), *Smarca4* (50%), *Hif1a* (400%), and *Stat3* (350%) (*Figure 7*).

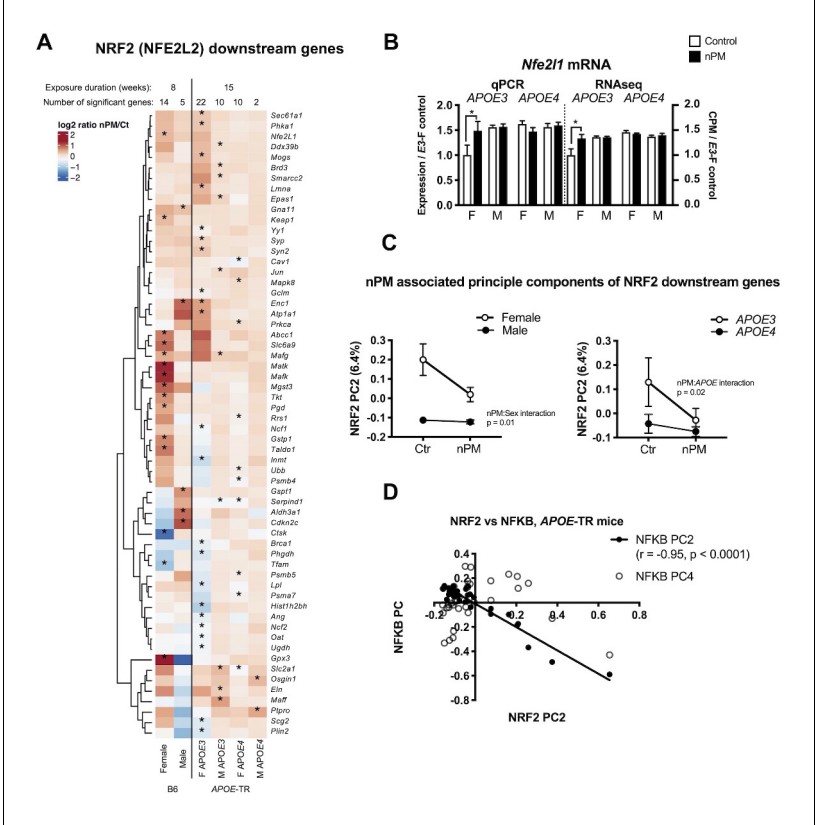

**Figure 5.** *Nrf2* responses to nPM in B6 and *APOE*-TR mice. (**A**) Heatmap of log2 fold changes of nPM responses, showing altered expression of at least 60 genes downstream of *Nrf2*, differing by sex or *APOE* genotype. (**B**) Validation by qPCR of *Nfe2l1* (*Nrf1*) changes in RNAseq. (**C**) Principal component analysis of 513 *Nrf2* downstream genes in *APOE*-TR: Only PC2 (6.4% of variance) had nPM-sex interaction (p=0.01) and *APOE* (p=0.02). *APOE3* females had the highest nPM response. (**D**) PC2 of *Nrf2* downstream genes varied inversely (R2 = 0.91, p=0.0001) with the PC2 of *Nfkb* downstream genes. Sample size of 4/genotype/sex/treatment.

## Discussion

These finding address the gap in how sex and *APOE* allele interactions may alter neurodegenerative responses to air pollution. Cerebral cortex genomic responses to nPM, were examined in C57BL/6 mice (B6, wildtype) and B6 mice carrying human transgenes for *APOE*3 and *APOE*4 alleles. Female mice had two-fold more genes that responded to nPM, further enhanced for *APOE*3. Responding genes included neuronal pathways (e.g. axonal guidance; glutamate synapse genes), inflammation (e.g. AMPK, and APK/JNK signaling), and antioxidant and hypoxic signaling (e.g. NRF2, HIF1A signaling). Genes in pathways downstream of NFKB and NRF2 responded oppositely to nPM. These interactions of NRF2 and NFKB may modulate sex and *APOE* risk for AD and accelerated cognitive aging during air pollution exposure.

The nPM responding genes may help to identify GxE in neurodegenerative risks from air pollution and cigarette exposure. For example, a combination of air pollution and a specific *IL1B* variant increased the risk of Parkinson disease (*Lee et al., 2016*). Our findings extend microarray analysis of frontal cortex of children and young adults from Mexican cities differing in air pollution levels: the 134 responding genes include inflammation (e.g. *NFKB*) and antioxidant responses (e.g. *GPX2*, *GPX3*) (*Calderón-Garcidueñas et al., 2012*). Microarray analysis of rat brain chronically exposed to PM0.2 also overlapped with our results: *S100a9*, calcium channels (e.g. *Cacna1i*), and glutamate receptors (*Ljubimova et al., 2013*).

These glutamatergic gene responses extend findings that hippocampal neurites are selectively sensitive to nPM (*Morgan et al., 2011*; *Woodward et al., 2017b*; *Fonken et al., 2011*). The increased levels of the ionotropic receptor NMDA type subunit 1 (*Grin1*) is notable: a *GRIN*

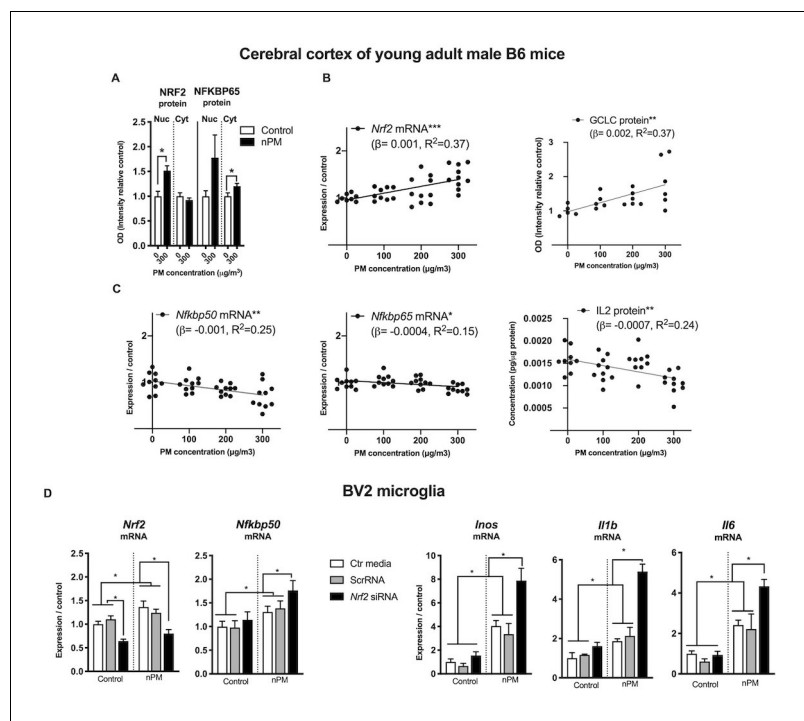

**Figure 6.** NRF2 and NFKB interact with nPM toxicity in cerebral cortex of male C57BL/6 mice and in mouse microglia (BV2 cells, in vitro). (**A**) Increased nuclear translocation of NRF2 and cytosolic NFKBP65 of B6 mice exposed to 300 µg/m$^3$ nPM for three wks. (**B**) nPM exposure dose-dependent increase of *Nrf2* mRNA and positive correlation with increase of GCLC protein. (**C**) nPM dose-dependent decrease of *Nfkbp65* and *Nfkbp50* mRNA, and IL2 protein levels. Inhalation exposure to nPM at 100, 200, and 300 µg/m$^3$ nPM (in vivo sample size, 10/group; exposure, 5 hr/d, 3 d/wk, 3 wks. **p=0.001, ***p=0.0001. (**D**) BV2 microglia in vitro response to nPM at 5 µg/ml nPM for 6 hr after partial knockdown of *Nrf2* (sample size, 6/group; two independent biological replicates). *Nrf2* mRNA knockdown was >60% at time 0. ANOVA with FDR multiple test correction. Mean ± SEM. *Adj. p=0.05. nPM chemical characterization (*Figure 1—figure supplement 1*).

polymorphism is associated with the risks of Parkinson (*Wu et al., 2010*), schizophrenia (*Demontis et al., 2011*), and also interacts with *APOE* alleles for earlier AD onset (*de Oliveira et al., 2016*), *GRIN* variants might also alter air pollution neurodegenerative responses, for example mutations in *GRIN2A* and *GRIN2B* increased the risk of cognitive impairments for lead poisoning of children (*Rooney et al., 2018b*). The nPM responses of *Grin2a* and *Grin2b* mRNA, while modest (p=0.05–0.06), merit further study among xenobiotics.

AD-associated genes responded differently to nPM by sex and *APOE* allele. *APOE3* females had the lowest baseline and the highest nPM response in genes associated with amyloidogenesis and TAU. The smaller responses of *APOE4* mice to nPM may be a ceiling effect, because about 7% of the nPM responsive genes also had *APOE4* baseline differences in both sexes. For non-exposed controls (baseline), *APOE4* vs *APOE3* differed in 300 genes related to known *APOE* pathways including LXR/RXR (*Courtney and Landreth, 2016*), Atherosclerosis (*Mahley et al., 2009*), and Rheumatoid arthritis (*Toms et al., 2012*). A similar *APOE* allele specificity was found in responses to ozone (O$_3$), which impaired memory in *APOE3*, but not *APOE4*-TR mice (*Jiang et al., 2019*). This pattern may be compared with ceiling effects of aging on responses to air pollution which we observed in middle-aged B6 mice. Both sexes at age 18 months have elevated baseline levels of antioxidant and inflammatory gene mRNA and protein, which did not respond to nPM or O3, unlike young mice (*Jiang et al., 2019*; *Woodward et al., 2017b*; *Zhang et al., 2017a*).

Brain amyloid must also be considered in the complex interactions of APOE genotype and sex with air pollution. The current study of *APOE*-TR mice response to nPM and a study of response to ozone (*Jiang et al., 2019*) showed that neither air pollutant altered brain Abeta40 and Abeta42 peptide levels for mice of ages 3 to 18 months (young to middle-age). The *APOE*-TR mice have the

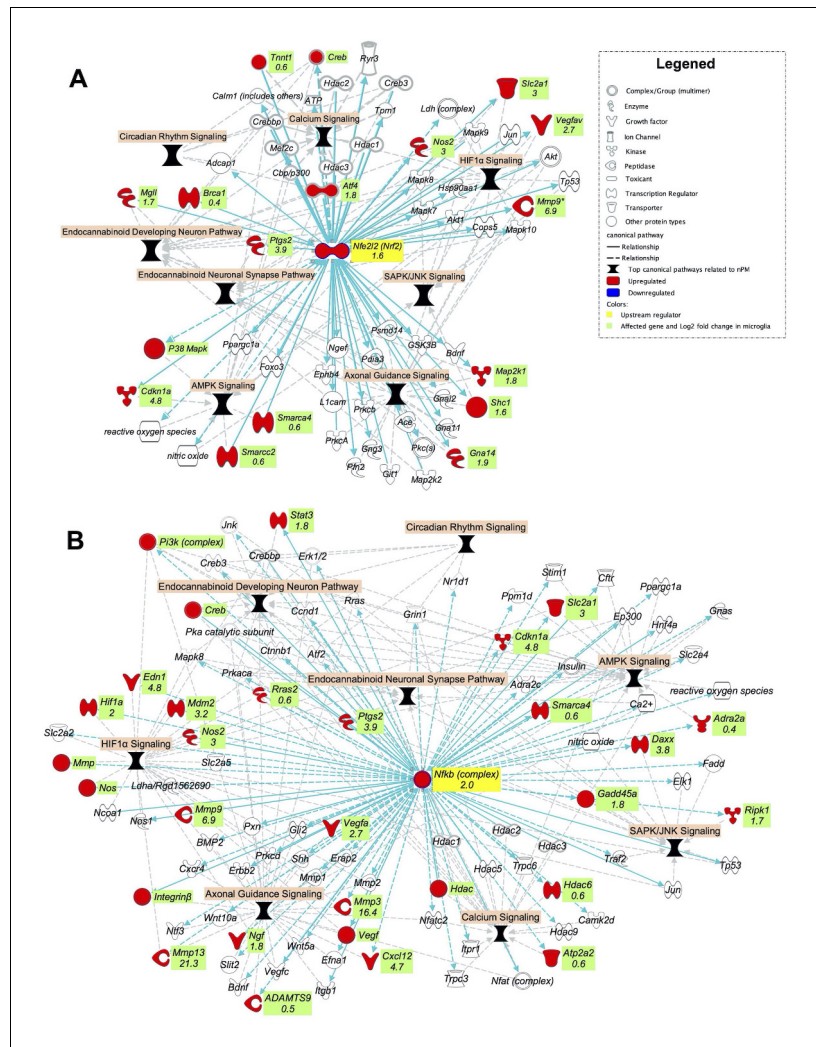

**Figure 7.** NRF2 and NFKB are potential upstream regulators of the top canonical pathways related to nPM effects in the cerebral cortex of adult mice. Gene networks of (**A**) NRF2 and (**B**) *NFKB* downstream genes in the top nPM related canonical pathway (*Figure 1A*). The network was made by IPA software. The networks were overlaid with the significant responses to nPM in mixed glial culture (*Woodward et al., 2017a*). The numbers indicate log2 fold-changes of gene expression in Affymetrix microarray. Dataset from prior studies (*Woodward et al., 2017a*). In vitro sample size, 4/group.

normal murine wildtype amyloid peptide Abeta42, which differs from the human in three amino acids and aggregates less avidly (*Otvos et al., 1993*). Thus, our study in *APOE*-TR mice do not include additional stress due to amyloid aggregates present in the *FAD* transgenic models. Unlike *APOE*-TR, nPM exposure increased the amyloid plaque loads in EFAD (*APOE*-TR plus five mutation in amyloidogenic genes) and *J20* AD mouse models (*Cacciottolo et al., 2017*; *Cacciottolo et al., 2020*). In EFAD mice, nPM caused greater amyloid plaque load in E4FAD than E3FAD. Even in that study, nPM increased levels of Abeta oligomers, but not plaque load in E3FAD more than for E4FAD mice. These differences suggest the hypothesis that *APOE* alleles cause differences in amyloid clearance, phagocytosis, and proteolysis. As expected, some nPM-responding genes that mediate amyloid clearance differed by the *APOE* allele. The complement factor C3, which mediates amyloid clearance (*Maier et al., 2008*), had a higher baseline in *APOE*3, but lacked nPM response, whereas *Vav* and S100 calcium-binding protein A9 (*S100a9*) responses to nPM were restricted to *APOE*3. *Vav* and *S100a9* regulate phagocytosis of fibrillary Abeta by microglia (*Wilkinson et al., 2006*; *Kummer et al., 2012*). The *APOE*3 and 4 proteins, and Abeta compete differentially for uptake by astrocytes via the lipoprotein receptor-related protein 1 (LRP1)-dependent pathway

(*Verghese et al., 2013*). Moreover, *APOE4* astrocytes have less efficient Abeta clearance, attributed to acidification of endosomes (*Verghese et al., 2013*; *Prasad and Rao, 2018*). Possibly, an efficient clearing system could compensate even for a greater amyloidogenic effect of nPM in *APOE3* than *APOE4* mice. This hypothesis will be tested further in vitro and in vivo.

Similar to the amyloid pathway, *APOE4* had higher baseline level of inflammation, while their response to nPM was less than the *APOE3*. This allele specificity was also shown for responses to ozone (*Jiang et al., 2019*). In contrast, responses to LPS endotoxin by intra-peritoneal injection were greater in *APOE4* male mice for microglial activation and systemic inflammatory cytokines (*Zhu et al., 2012*). In vitro, *APOE4* macrophages also had higher induction of NFKB, TNFA, and heme oxygenase one in response to LPS (*Jofre-Monseny et al., 2007*). While air pollution can include endotoxins, the comparisons with injected LPS for sex and *APOE* are limited because the systemic responses are downstream of the lung, which receives most inhalants.

Female mice had greater response to nPM than males for immune and antioxidant pathways, for wildtype and *APOE*-TR mice. Sex hormones and early life gonadal programming during brain development could underly these differences. Ex-vivo microglial cultures from male and female brains had divergent inflammatory response to estradiol (E2) and LPS (*Loram et al., 2012*). A gene expression microglial developmental index showed a sex difference in maturation and immune reactivity, which correlated with the risk of AD and autism spectrum disorders (*Hanamsagar et al., 2017*). In SH-SY5Y neuroblastoma cells, E2 increased cell survival and NRF2 antioxidant defense against homocysteine (*Chen et al., 2013*). Similarly, E2 treatment of postnatal rats ameliorated acute ethanol-induced oxidative stress, neuroinflammation, and neuronal cell death through increase of sirt1, P53 acetylation inhibition, and reduction in phospho-NFKB nuclear localization (*Khan et al., 2018*). Higher adaptive genomic response by females might favor faster detoxification or recovery from air pollution. Metformin mediated NRF2 activation could ameliorate tight junction proteins, blood–brain barrier (BBB) integrity, reduce inflammation and oxidative stress, and also normalize the levels of BBB glucose transporter GLUT1 protein after cigarette smoke exposure in mice (*Prasad et al., 2017*).

As opposed to potential protective effects of estrogen hormones against air pollution, female mice had higher baseline levels of Abeta40 than males. This results parallels with female excess amyloid plaque load of aged AD mouse models (*Callahan et al., 2001*; *Hirata-Fukae et al., 2008*). Neonatally demasculinized or defeminized *3xTg*-AD or other AD transgenic mice shows the major role of sex steroids in determining adult sex differences in Abeta accumulation (*Carroll et al., 2010*; *Pike et al., 2009*). Thus, sex steroids can act as a double-edged sword for amyloidogenic responses to air pollution and other environmental neurotoxins. Resolving the role of steroid hormones in this intricate relationship of background biology and air pollution requires studies on the recovery after nPM exposure in gonadectomized and older mice. We plan further studies of the complex interface of sex-*APOE* allele and nPM in mixed glial cultures derived from male and female *APOE3* and *APOE4*-TR mice.

NFKB and NRF2 had opposite responses to nPM, that included downstream genes in wildtype B6 and *APOE*-TR. This divergence was also shown in a short term (3 weeks) exposure of B6 male mice. The NRF2 and NFKB crosstalk was validated in BV2 microglia. This is the first evidence for NRF2 and NFKB interactions in response to air pollution of both in vivo and in vitro models. These results parallel the LPS responses of monocytes, which showed redox-mediated transcriptional cross-talk between NRF2 and NFKB responses to LPS (*Zhang et al., 2017b*). Concurrent increase of nuclear NRF2 and cytosolic NFKBP65 in cerebral cortex after nPM exposure suggest that NRF2 activation can attenuate NFKB nuclear localization. We hypothesize involvement of KEAP1, the NRF2 repressing protein, which can mediate IKKB degradation and inhibit NFKB nuclear localization (*Kim et al., 2010*; *Lee et al., 2009*). Other mechanisms could be mediated by direct protein-protein interaction, and by secondary messengers. For example, NRF2 can inhibit NFKB through reduction of reactive species and suppress RAC1-mediated NFKB activation (*Sanlioglu et al., 2001*; *Cuadrado et al., 2014*). In contrast, NFKB can also inhibit NRF2 activity through enhancing the recruitment of histone deacetylase (HDAC3) to ARE region (*Wakabayashi et al., 2010*), or competing with NRF2 for binding to CH1-KIX domain of CBP protein inside the nucleus (*Liu et al., 2008*). These interactions are also shown for thein nematode *C. elegans:* the *NRF2* homologue *skn-1* and the antibacterial factor 2 (*abf-2*) responded rapidly to nPM, with persisting developmental effects (*Haghani et al., 2019*).

Since immortalized BV2 microglial cells have limited comparability with in vivo microglial cells, we further corroborated the nPM responses of adult brain in primary mixed glial culture. In adult brain, nPM affected a network of genes from a set of highly interconnected canonical pathways including NRF2 oxidative stress response, HIF1A, AMPK, circadian rhythm, and endocannabinoid related signaling pathways. Interestingly, numerous genes from this network including *Nrf2* and *Nfkb* mRNA were upregulated in mixed glial culture after 24 hr acute exposure to nPM. A large portion of this gene network were considered as downstream of NRF2 and NFKB transcriptional factors. Thus, these two molecules are potentially the hub upstream regulators of long-term nPM neurotoxic effects. This hypothesis remained to be tested in further in vitro and in vivo studies.

The statistical power to identify all responding genes is intrinsically limited by the high dimensionality of RNAseq data with >20,000 genes. Thus, p=0.05 threshold needs to be adjusted for multiple testing to identify the real changes. Unfortunately, the small effect size of nPM exposure together with the necessarily small sample size of animals led to lack of enogh power to detect the changes at 5% FDR rate. To minimize the rate of false positives, we reported the changes at a nominal significance of p<0.005. Nonetheless, there could be small changes in some genes that we could not detect but still critical for air pollution toxicity. For example, while RNAseq analysis did not detect responses of *Nrf2*, *Nfkb*, and *Gclc* mRNA at p<0.005, the nPM dose-response experiment showed a 50% dose-dependent change in *Nrf2*, *Nfkb* mRNA, and GCLC protein, which confirmed findings on the cerebellum (*Zhang et al., 2012*). Thus, RNAseq and other high dimensional data are inherent with potential false-positive and false-negative results.

Another limitation of air pollution studies is the heterogeneity in the chemical composition and toxic activity PM. Recently, we identified physical and chemical characteristics of nPM that altered in vitro and in vivo toxicity of nPM (*Zhang et al., 2019*; *Haghani et al., 2020*). Regardless, NRF2 and NFKB responses were consistent in all the nPM batches used in these different experiments.

In summary, air pollution neurotoxicity was shown to have sex- and *APOE* allele-specificity, which are main risk factors for AD. These findings give a rationale for including *APOE*-gender interactions in epidemiological studies of cognitive aging and dementias.

# Materials and methods

## Key resources table

| Reagent type (species) or resource | Designation | Source or reference | Identifiers | Additional information |
|---|---|---|---|---|
| Genetic reagent (*M. musculus*) | *APOE3*-TR$^{+/+}$ *APOE4*-TR$^{+/+}$ | PMID:8980023 | | |
| Strain, strain background (*M. musculus*) | C57BL/6J (B6) | Jackson laboratory | 000664; RRID:IMSR_JAX:000664 | |
| Cell line (*M. musculus*) | BV2 microglia | ATCC | EOC 20 (ATCC CRL-2469); RRID:CVCL_5745 | Female originated |
| Other | Mixed glia (microglia and astrocyte) | *R. norvegicus* | | Postnatal days 3–5, mixed sexes |
| Transfected construct (*M. musculus*) | *Nfe2l2* siRNA | Thermofisher Scientific | 156499 | |
| Other | Lipofectamine RNAiMAX reagent | Thermofisher Scientific | 13778500 | |
| Antibody | anti-NRF2 (rabbit polyclonal) | Abcam | ab137550; RRID:AB_2687540 | WB, 1:1000 |
| Antibody | anti-H3 (rabbit polyclonal) | Cell Signaling Technology | D1H2; RRID:AB_10544537 | WB, 1:1000 |
| Antibody | anti-GAPDH (Mouse monoclonal) | Santa Cruz Biotechnology | sc-32233; RRID:AB_627679 | WB, 1:500 |

*Continued on next page*

*Continued*

| Reagent type (species) or resource | Designation | Source or reference | Identifiers | Additional information |
|---|---|---|---|---|
| Antibody | anti-NFKBP65 (Rabbit polyclonal) | Cell Signaling Technology | D14E12; RRID:AB_10859369 | WB, 1:750 |
| Antibody | anti-mouse IRDye 800CW | LICOR | 926–32210; RRID:AB_621842 | WB, 1:20,000 |
| Antibody | anti-rabbit IRDye 680RD | LICOR | 926–68070; RRID:AB_10956588 | WB, 1:20,000 |
| Commercial assay or kit | RNAeasy Mini Kit | Qiagen | 74104 | |
| Commercial assay or kit | TRUseq Stranded mRNA Kit | Illumina | 20020594 | |
| Commercial assay or kit | qScript cDNA Supermix | Quantabio | | |
| Commercial assay or kit | Taq master mix | Biopioneer | MAT-2.1–10 | |
| Commercial assay or kit | 12–230 kDa Jess or Wes Separation Module | Protein Simple | SM-W004 | |
| Commercial assay or kit | V-PLEX proinflammatory panel one immunoassay | Mesoscale Diagnostics, Rockville, MD | K15048D | |
| Commercial assay or kit | V-PLEX Aβ Peptide Panel 1 (4G8) Kit | Mesoscale Diagnostics, Rockville, MD | K15199E | |
| Chemical compound, drug | TRIzol | Invitrogen | 15596026 | |
| Software, algorithm | Rstudio | | | Packages: LIMMA, WGCNA, BRETIGEA |
| Software, algorithm | Ingenuity pathway analysis | Qiagen | | |
| Software, algorithm | GraphPad Prism | | | Version 8 |

## Animals

Husbandry and experimental procedures were approved by the USC Institutional Animal Care and Use Committee (approval numbers: *APOE*-TR experiment, 20417; B6 experiments and mixed glial culture, 11233). The C57BL/6J and *APOE*-TR (*Xu et al., 1996*) mice were aged 2 months at exposure onset (*Cacciottolo et al., 2016*; *Youmans et al., 2012*). For long-term nPM exposures (8–15 weeks), four mice for each sex, genotype (C57BL/6J, *APOE*3-TR, *APOE*4-TR), and exposure (48 mice total) were randomly assigned to nPM exposure or control. The dose-response experiment was done with 10 male C57BL/6J mice per group. After exposure, mice were euthanized by isoflurane anesthesia and perfused transcardially with phosphate-buffered saline. Brains were hemisected at midline; total cerebral cortex was frozen on dry ice and stored at −80°C. Investigators were blinded to exposure groups during data measurement and analyses.

## Air pollution nPM collection and exposure

Mice were exposed to nPM, a nano-sized subfraction of airP particulate matter of 2.5 microns diameter (PM2.5) collected from a local urban freeway corridor (*Woodward et al., 2017b*; *Haghani et al., 2020*). Briefly, PM0.2 samples were collected by a High-Volume Ultrafine Particle (HVUP) Sampler (*Misra et al., 2002*) at 400 L/min flow rate on an 8 × 10 inch-Zeflour PTFE filter (Pall Life Sciences, Ann Arbor, MI). The Particle Instrumentation Unit of University of Southern California is located within 150 m downwind of a major freeway (I-110). Chemical composition and size distribution of re-aerosolized nPM is characterized by high-resolution mass spectrometry (SF-ICPMS) and Sievers 900 Total Organic Carbon Analyzer as described before (*Morgan et al., 2011*; *Haghani et al., 2020*). Chemical characterization of the nPM batches in this study is in Supplementary data (*Figure 1—figure supplement 4*). Re-aerosolized nPM or filtered air (control) was

delivered to the sealed exposure chambers at approximately 300 µg/m3 concentration to model chronic exposure: 5 h/day, 3 days/week, for 8 (C57BL/6J) or 15 weeks (*APOE*-TR). For dose-response experiment, 8 weeks male C57BL/6J mice were exposed to approximately 100, 200, and 300 µg/m$^3$ for 3 weeks. The duration and nPM dosages were based on brain responses in prior studies (*Haghani et al., 2020*; *Cheng et al., 2016*). The 3 weeks of intermittent exposure (5 h/day, 3 days/week) to 300 µg/m$^3$ nPM yields an average hourly exposure of 27 µg/m$^3$, as experienced in many cities.

## RNA sequence (RNAseq) analysis of mouse cerebral cortex

RNA was extracted with TRIzol (Invitrogen) and RNAeasy Mini Kit (Qiagen) with DNase digestion. Libraries were made with the TRUseq Stranded mRNA Kit (Illumina) with 1 mg of RNA. For Illumina NextSeq500sequencing, a single end-sequencing length of $\geq$50 nt was used. Reads were aligned and quantified to the mouse reference genome RefSeq mm10 with Tophat2 (v2.0.8b), restricted to uniquely mapping reads with three possible mismatches using the Partek flow software platform (*Trapnell et al., 2012*).

Differential gene expression was calculated by linear modeling (Limma package in R). Significance was calculated at 5% FDR rate or p-value, 0.005. Gene responses were analyzed by Qiagen Ingenuity Pathway Analysis (IPA) software. In combining the datasets generated from B6 and *APOE*-TR experiments, the models were adjusted by the COMBAT method to control for unknown variance (*Johnson et al., 2007*). Additional downstream analysis and plotting were done in Rstudio and GraphPad Prism. Cell type deconvolution analysis was done using BRETIGEA (BRain cEll Type spe-cIfic Gene Expression Analysis) R package (*McKenzie et al., 2018*), which uses single-cell RNAseq data to identify cell-type-specific gene-signatures to predict the proportion of each cell type in bulk RNAseq data.

## Weighted gene co-expression network analysis (WGCNA)

The co-expression network, based on WGCNA, was constructed from RNAseq data. WGCNA is an unsupervised clustering approach, which assigns groups of genes with shared expression patterns into modules (*Langfelder and Horvath, 2008*). Briefly, the adjacency matrix (correlations between genes) was converted to a scale-free network using soft threshold power (tuned in each group) of the signed matrix. The result was converted to a topological overlap matrix (TOM). Hierarchical clustering used 1-TOM distance measure (dissimilarity). A dynamic tree-cut algorithm was used to assign modules containing at least 30 genes. Module eigengenes (MEs) were calculated as the maximum amount of the variance of the model for each module, based on the Singular Value Decomposition method. Linear regression models estimated association of nPM or *APOE* with the MEs. The top 150 hub genes of the modules were selected for IPA analysis by the highest eigengene connectivity (kME). In total, 32 nPM associated gene modules were identified based on different analyses. Thus, the modules were renamed to M1-M32 to distinguish each analysis.

## Cell culture and *Nrf2* siRNA

BV2 microglia (mouse-derived) were grown in Dulbecco's modified Eagle's medium (DMEM)/F12 (Cellgro, Mediatech, Herndon, VA) containing 10% fetal bovine serum, 1% penicillin/streptomycin, and 1% L-glutamine (*Woodward et al., 2017a*). These cells were authenticated by expression of microglial markers, cell morphology, phagocytic activity, and comparison of nPM or LPS responses with mixed glial culture and other literature. The cells were not tested for mycoplasma. *Nfe2l2* (*Nrf2*) siRNA (156499, Thermofisher Scientific) was delivered by Lipofectamine RNAiMAX reagent (Thermo-fisher Scientific).

## Quantitative real-time PCR

Extracted RNA was reverse transcribed to cDNA using qScript cDNA Supermix (Quantabio). qPCR used Taq master mix (Biopioneer) and gene-specific primers (*supplementary file 1*; *Figure 1—figure supplement 4*).

## Protein extraction

Frontal cerebral cortex (anterior to Bregma, excluding olfactory bulbs) was homogenized (20 mg in 0.2 ml) in 1x RIPA buffer supplemented with 1 mM $Na_3VO_2$, 1 mM phenylmethane sulfonyl fluoride (PMSF), 10 mM NaF, phosphatase inhibitor cocktail (Sigma), and Complete Mini EDTA-free Protease Inhibitor Cocktail Tablet (Roche). Supernatants were obtained by centrifugation at 12,000 g/15 min.

## NRF2 subcellular localization

Nuclear and cytosolic fractions were separated after tissue homogenization in sucrose-Tris-$MgCl_2$ (STM) buffer with phosphatase and protease inhibitors and centrifuged 800g $\times$ 15 min (Dimauro et al., 2012). After removing supernatant, the nuclear pellet was washed in STM buffer, resuspended in HEPES pH 7.9 buffer (20 mM HEPES 1.5 mM $MgCl_2$, 0.5 M NaCl, 0.2 mM EDTA, 20% glycerol, 1% Triton-X-100, protease and phosphatase inhibitors) and sonicated. Cell fraction purity was validated by immunoblotting for nuclear histone 3 (H3) and cytosolic glyceraldehyde 3-phosphate dehydrogenase (GAPDH).

## Protein analysis

NRF2 was detected by Western blot using anti-NRF2 primary antibody (1:1000, rabbit polyclonal, ab137550). Proteins (20 µg) were electrophoresed on Criterion 4–15% TGX gels (Biorad) and transferred to PVDF membranes. After washing with TBS+0.05% Tween-20 (PBST), membranes were blocked (LiCOR) 1 hr/ambient, then incubated with primary antibody overnight at 4°C: anti-NRF2 (1:1000, rabbit polyclonal, ab137550), anti-NFKBP65 (1:750, Rabbit polyclonal, Cell Signaling Technology, D14E12), anti-H3 (1:1000, Rabbit polyclonal, Cell Signaling Technology, D1H2), and anti-GAPDH (1:500, Mouse monoclonal, Santa Cruz Biotechnology, sc-32233). Bands were identified by incubation with 1:20,000 fluorochome-conjugated LICOR-antibodies (anti-mouse IRDye 800CW or anti-rabbit IRDye 680RD); band intensity was analyzed by ImageJ. GCLC was assayed by capillary electrophoresis (12–230 kDa range, Jess ProteinSimple, California, USA). Total lysate 1 µg/µl was electrophoresed and treated with anti-GCLC (1:100) and HRP-labeled secondary antibody. Results were normalized to total protein (PN module, ProteinSimple). IL2 was assayed by V-PLEX proinflammatory panel one immunoassay (Mesoscale Diagnostics, Rockville, MD). Abeta 40 and 42 peptides were assayed by 4G8 Kit VPLEX (Peptide Panel 1, Meso Scale Discovery, Rockville, MD).

## Acknowledgements

We are grateful to Drs. Arian Saffari and Farimah Shirmohammadi (USC Viterbi School of Engineering) for their contribution to nPM collection and animal exposures.

## Additional information

### Funding

| Funder | Grant reference number | Author |
| --- | --- | --- |
| Cure Alzheimer's Fund | | Caleb Finch |
| National Institute on Aging | R01-AG051521 | Caleb Finch |
| National Institute on Aging | P50-AG05142 | Caleb Finch |
| National Institute on Aging | P01-AG055367 | Caleb Finch |
| National Institute on Aging | T32- AG052374 | Amin Haghani |
| National Institute on Aging | T32-AG000037 | Max Thorwald |
| National Institute on Aging | 1RF1AG053982-01A1 | Terrence Town |
| National Institute on Aging | 1R01AG057912-01 | Morgan Levine |
| National Institute on Aging | 4R00AG052604-02 | Morgan Levine |

The funders had no role in study design, data collection and interpretation, or the decision to submit the work for publication.

## Author contributions
Amin Haghani, Conceptualization, Data curation, Software, Formal analysis, Investigation, Visualization, Methodology, Writing - original draft, Writing - review and editing; Mafalda Cacciottolo, Investigation, Methodology; Kevin R Doty, Data curation; Carla D'Agostino, Max Thorwald, Nikoo Safi, Hongqiao Zhang, Investigation; Morgan E Levine, Methodology; Constantinos Sioutas, Resources, Writing - review and editing; Terrence C Town, Resources, Funding acquisition; Henry Jay Forman, Writing - review and editing; Todd E Morgan, Conceptualization, Resources, Project administration, Writing - review and editing; Caleb E Finch, Conceptualization, Supervision, Funding acquisition, Writing - review and editing

## Author ORCIDs
Amin Haghani ⬦ https://orcid.org/0000-0002-6052-8793
Morgan E Levine ⬦ https://orcid.org/0000-0001-9890-9324
Caleb E Finch ⬦ https://orcid.org/0000-0002-7617-3958

## Ethics
Animal experimentation: All husbandry and experimental procedures were approved by the USC Institutional Animal Care and Use Committee. Approval numbers: APOE-TR experiment, 20417; B6 experiments and mixed glial culture, 11233.

## Decision letter and Author response
Decision letter https://doi.org/10.7554/eLife.54822.sa1
Author response https://doi.org/10.7554/eLife.54822.sa2

# Additional files

## Supplementary files
- Source code 1. Differential expression analysis of combined wildtype and *APOE*-TR datasets.
- Source code 2. Differential expression analysis of female *APOE3* cerebral cortex.
- Source code 3. WGCNA in female *APOE3* animals.
- Supplementary file 1. Specific primers used in this study.
- Supplementary file 2. Excel file with the results of different models in this study.
- Transparent reporting form

## Data availability
All raw data have been deposited in GEO under accession code GSE142066.

The following dataset was generated:

| Author(s) | Year | Dataset title | Dataset URL | Database and Identifier |
|---|---|---|---|---|
| Haghani A, Cacciottolo M, Doty KR, D'agostina C, Thorwald M, Safi N, Saffari A, Shirmohammadi F, Levine ME, Sioutas C, Town TC, Forman HJ, Zhang H, Morgan TE, Finch CE | 2020 | Mouse brain transcriptome responses to inhaled nanoparticulate matter differed by sex and APOE in Nrf2-Nfkb interactions | https://www.ncbi.nlm.nih.gov/geo/query/acc.cgi?acc=GSE142066 | NCBI Gene Expression Omnibus, GSE142066 |

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
