## [Decision Letter]

**Acceptance summary:**

This interesting study addresses pressing questions about the interactions between sex and *ApoE* in response to nPM. The authors' RNAseq data demonstrate various sex differences in response to nPM, as well as several genotype differences in response to nPM. The authors also show data suggesting that there is an interaction between *Nrf2* and *NFkB* in the nPM response.

**Decision letter after peer review:**

Thank you for submitting your article "Mouse brain transcriptome responses to inhaled nanoparticulate matter differed by sex and *ApoE* in *Nrf2-NFkB* interactions" for consideration by *eLife*. Your article has been reviewed by three peer reviewers, and the evaluation has been overseen by K VijayRaghavan as the Senior and Reviewing Editor. The following individual involved in the review of your submission has agreed to reveal their identity: Riu-Ming Liu (Reviewer #1).

The reviewers have discussed the reviews with one another and the Reviewing Editor has drafted this decision to help you prepare a revised submission.

We would like to draw your attention to changes in our revision policy that we have made in response to COVID-19 (https://elifesciences.org/articles/57162). Specifically, when editors judge that a submitted work as a whole belongs in *eLife* but that some conclusions require a modest amount of additional new data, as they do with your paper, we are asking that the manuscript be revised to either limit claims to those supported by data in hand or to explicitly state that the relevant conclusions require additional supporting data.

Summary:

The etiology of Alzheimer's disease is unclear. This study explores the potential interactions between a genetic risk factor (*APOEe4*) and environmental exposure (particulate matter exposure) in the regulation of gene expression in the brain and sex effect, using human *ApoE* 3 and 4 replacement mice. Transcriptome responses to inhaled nanoparticulate matter were determined by RNAseq and partially confirmed by qPCR, using mouse brain tissues from wild type and *ApoE3* or 4 mice. The transcriptome data were analyzed by Weighted Gene Co-Expression Network Analysis (WGCNA) and Ingenuity Pathway Analysis (IPA) etc. software. Crosstalk between *Nrf2* and *NFkb* pathways was further studied using culture microglial cells. The study is significant and is innovative, as no such study has been reported yet. The methods used to analyze gene expression are standard and reliable. The results are interesting, although not completely surprising. However, some issues remain to be addressed.

Specific concerns:

1) No information about the sample size is provided in any of the figures. It is unclear whether the transcriptome data are from one single run or whether the results have been repeated and how many samples have been used in each group.

2) FDR values will be more convincing than p-values for analysis of RNAseq data. It is unclear why the p-value was set to 0.005, instead of 0.05, for DEGs.

3) Figure 1 and 2, what M1…M32 stands for needs to be specified clearly.

4) Figure 1 legend is messed up. Should C be D, D be E and E be F? The labels should be revised in the text as well.

5) Figure 1—figure supplement 3, panel B. Materials and methods section should be detailed about how cell type deconvolution analysis of RNAseq was conducted. What genes have been used to identify these cells?

6) Figure 3A shows clearly a sex and *ApoE* genotype dependent response to nPM in amyloid peptide (Ab) metabolism (synthesis and clearance). In *ApoE3* mice, males have higher expression levels of the genes involved in Ab synthesis than female mice under untreated condition. These data seem contradictory to published data, which show more Ab in females than in male. It is also interesting to notice that *ApoE4* mice express less Ab clearance related genes than *ApoE3* mice (C3 and C3ar1). It is strongly suggested to measure Ab levels in these mice to correlate to these gene expression profiles with Ab levels.

7) Only female E3, not female E4, responded to nPM in Ab pathway genes. The potential mechanism need to be discussed.

8)Figure 4. PC2 and PC4 need to be defined.

9) In general, E3 mice, both male and female, responded more drastically than E4 mice to nPM in *Nrf2* and *NFkb* pathways as well as Ab metabolism. These findings seem to be unexpected, based on the literature about *ApoE3* and *ApoE4* mice. What is the potential mechanism and biological significance needs to be discussed in details.

10) The paper would be greatly strengthened by including a dose-response in female B6 mice, and performing in vitro *Nrf2*/*NFkB* experiments in microglia from female mice. As much of the paper focuses on sex differences in response to nPM, and Figure 4 shows clear sex differences in NF-κB responses to nPM, these should be extended to the experiments described in Figure 6. Ideally, the in vitro experiments should be repeated with primary microglia from male and female mice, rather than BV2 cells. This would also substantially increase the strength of the model proposed in the graphical abstract. Minimally, the authors need to address the caveats of the BV2 experiments in the Discussion.

11) A rationale for the doses selected in the dose-response should be included in the paper; the lowest selected dose of 100ug/m3 nPM is still very high relative to typical human exposures. While it may not be feasible to repeat the dose-response studies, a more informative and typical dose response would likely include doses that differ by an order of magnitude i.e. 1, 10 and 100 μg/m3.

12) The logic for focusing on *Nrf2* pathways based on the RNAseq data is not clear. Based on the IPA analysis shown in Figure 1B, the NRF-2 pathway appears much less affected than calcium, HIF1a, or AMPK signaling. While this may reflect the fact that *ApoE3*-TR mice did not appear to respond via Nfe2L2, it is somewhat surprising that the canonical *Nrf2*-mediated oxidative stress response pathway is implicated in the multivariate model (Figure 1B) but not the stratified pathway analysis (1F). Although there may be valid reasons to investigate NRF2 based on previous studies (as described in the Introduction), the RNAseq data in Figure 1 does not make a strong case for focusing on this pathway specifically. Likewise, it is not clear why genes identified by the RNAseq study (i.e. Grin1) are not investigated further. Are these genes downstream of *NFkB* or *Nrf2* in the in vitro microglia models?

13) It is very interesting that amyloid deposition pathways were affected by nPM only in *ApoE* females. However, this would seem surprising, given that the authors previously saw increased amyloid deposition in E4FAD mice exposed to nPM compared to E3FAD mice. How do the authors reconcile the lack of AD-associated gene responses in female *ApoE4*-TR mice with their previous finding of increased amyloid deposition in female E4FAD mice exposed to nPM? This should be added to the Discussion.

14) The interaction chosen is striking: sex and *ApoE3/4* genotype. It seems one among many possible experimental designs to impose and is justified in the Introduction in a somewhat miscellaneous fashion.

15) About Abeta, only the *ApoE3* mice responded. Isn't that surprising?

16) Subsection “Air pollution nPM collection and exposure” need more detail about the chemical engineering used to prepare the samples. And "5 h/day, 3 days/week" for 8 weeks does not sound like "chronic exposure" to me.

17) Likewise, Results paragraph three: "Both analyses (DE, WGCNA) showed more nPM-responding gene responses for *ApoE3* than *E4*." Isn't that surprising?

18) In light of the senior author's long experience in genetics and molecular biology, it might be useful to use part of the Discussion to offer comments on the pathway analysis programs he uses: discussion of their epistomilogic status and the verity of the statistical analyses applied would head off charges of over-interpretation of the data presented here.

19) There seem to be emerging new theories re the chemical status of nanoparticle/microcavity relations. Can the authors use this in their Discussion?

---

## [Author Response]

Revisions for this paper:Specific concerns:1) No information about the sample size is provided in any of the figures. It is unclear whether the transcriptome data are from one single run or whether the results have been repeated and how many samples have been used in each group.

The sample sizes were added to all figure legends.

2) FDR values will be more convincing than p-values for analysis of RNAseq data. It is unclear why the p-value was set to 0.005, instead of 0.05, for DEGs.

Additional sentences were added to Discussion to explain why it is important to use a nominal significance level stronger than p 0.05. Discussion paragraph eleven.

3) Figure 1 and 2, what M1…M32 stands for needs to be specified clearly.

A sentence was added to WGCNA methods to define M1-M32 names.

4) Figure 1 legend is messed up. Should C be D, D be E and E be F? The labels should be revised in the text as well.

We apologize and corrected Figure 1 legend.

5) Figure 1—figure supplement 3, panel B. Materials and method section should be detailed about how cell type deconvolution analysis of RNAseq was conducted. What genes have been used to identify these cells?

We added details and references to Materials and methods subsection “RNA sequence (RNAseq) analysis of mouse cerebral cortex”. The markers are described in PMID29892006.

6) Figure 3A shows clearly a sex and ApoE genotype dependent response to nPM in amyloid peptide (Ab) metabolism (synthesis and clearance). In ApoE3 mice, males have higher expression levels of the genes involved in Ab synthesis than female mice under untreated condition. These data seem contradictory to published data, which show more Ab in females than in male. It is also interesting to notice that ApoE4 mice express less Ab clearance related genes than ApoE3 mice (C3 and C3ar1). It is strongly suggested to measure Ab levels in these mice to correlate to these gene expression profiles with Ab levels.

We welcomed this opportunity to measure amyloid peptides. We obtained new data for the cerebral cortex of APOE-TR animals. See Figure 3 and subsection “Baseline effects of ApoE4 allele and the overlap with nPM responses”.

7) Only female E3, not female E4, responded to nPM in Ab pathway genes. The potential mechanism need to be discussed.

See discussion of mechanisms in paragraph four and five of the Discussion section.

8) Figure 4. PC2 and PC4 need to be defined.

Principal components (PC) are defined in subsection “Sex- and ApoE- specific nPM mediated NFKB responses”.

9) In general, E3 mice, both male and female, responded more drastically than E4 mice to nPM in Nrf2 and NFkb pathways as well as Ab metabolism. These findings seem to be unexpected, based on the literature about ApoE3 and ApoE4 mice. What is the potential mechanism and biological significance needs to be discussed in details.

This result also surprised us. Further comparison with previous studies showed that this is a consistent pattern, Discussion paragraphs four and five. *ApoE4* allele elevated the baseline expression OF inflammatory and antioxidant genes. Thus, E4 MAY cause a ceiling effect in responsiveness to the environment. This is similar to the ceiling effect of age against nPM or O3 response. We added additional discussion points about this observation.

10) The paper would be greatly strengthened by including a dose-response in female B6 mice, and performing in vitro Nrf2/NFkB experiments in microglia from female mice. As much of the paper focuses on sex differences in response to nPM, and Figure 4 shows clear sex differences in NF-κB responses to nPM, these should be extended to the experiments described in Figure 6. Ideally, the in vitro experiments should be repeated with primary microglia from male and female mice, rather than BV2 cells. This would also substantially increase the strength of the model proposed in the graphical abstract. Minimally, the authors need to address the caveats of the BV2 experiments in the Discussion.

We appreciate the reviewers’ suggestion on studying sex-specific responses in mixed glial culture. We state our plans for a sex-specific mixed glial culture study, which will take us 6 to 9 months post COVID19. That said, we were able to honor the reviewers’ suggestion with analysis of microarray dataset that was available from a prior study. This new analysis adds depth by further documenting the importance of *Nrf2*, NF-κB transcriptional factors in initial responses of nPM.

11) A rationale for the doses selected in the dose-response should be included in the paper; the lowest selected dose of 100ug/m3 nPM is still very high relative to typical human exposures. While it may not be feasible to repeat the dose-response studies, a more informative and typical dose response would likely include doses that differ by an order of magnitude i.e. 1, 10 and 100 μg/m3.

During the past decade, our lab has tested a wide range of air pollution dosages in adult brains. The dosages were selected based on the observed neurotoxicity in our previous studies. these data suggest that the reviewer’s proposed concentrations would have minimal effect on adult mouse brain. The 300 μg/m3 is used by other labs in experimental studies (PMID30668980, PMID29523932, PMID27865893). These levels are experienced by many populations in Asia and Africa. In our model, exposure for 5hr/day, 3day/week) for 3 weeks yields an average hourly exposure of 27 μg/m3 that is experienced in any US cities. We added information to Materials and methods to justify these dosages.

12) The logic for focusing on Nrf2 pathways based on the RNAseq data is not clear. Based on the IPA analysis shown in Figure 1B, the NRF-2 pathway appears much less affected than calcium, HIF1a, or AMPK signaling. While this may reflect the fact that ApoE3-TR mice did not appear to respond via Nfe2L2, it is somewhat surprising that the canonical Nrf2-mediated oxidative stress response pathway is implicated in the multivariate model (Figure 1B) but not the stratified pathway analysis (1F). Although there may be valid reasons to investigate NRF2 based on previous studies (as described in the Introduction), the RNAseq data in Figure 1 does not make a strong case for focusing on this pathway specifically. Likewise, it is not clear why genes identified by the RNAseq study (i.e. Grin1) are not investigated further. Are these genes downstream of NFkB or Nrf2 in the in vitro microglia models?

We understand this concern. We added analysis in mixed glial culture transcriptome data to show the relationship of *NFkB* and *Nrf2* with other nPM related canonical pathways. Our new network analysis revealed that the top nPM related canonical pathway is highly interconnected. Moreover, *NFkB* and *Nrf2* are the upstream regulators of the numerous key regulators of all the highlighted pathways.

13) It is very interesting that amyloid deposition pathways were affected by nPM only in ApoE females. However, this would seem surprising, given that the authors previously saw increased amyloid deposition in E4FAD mice exposed to nPM compared to E3FAD mice. How do the authors reconcile the lack of AD-associated gene responses in female ApoE4-TR mice with their previous finding of increased amyloid deposition in female E4FAD mice exposed to nPM? This should be added to the Discussion.

We suggest that the reason behind higher Abeta plaque accumulation in E4FAD mice is the higher clearance capacity of the animals with the E3 genotype. Our study of EFAD female mice showed that nPM induced more Abeta oligomer formation in E3FAD than E4FAD. However, this pattern did not hold for amyloid plaques. As shown by PMID29946028, PMID23620513, E3 mice and cells have faster clearance of amyloid peptides. We expanded the topic in the Discussion paragraph four and five.

14) The interaction chosen is striking: sex and ApoE3/4 genotype. It seems one among many possible experimental designs to impose and is justified in the Introduction in a somewhat miscellaneous fashion.

Air pollution, sex, and *APOE* are among the major risk factors of late-onset of Alzheimer’s disease, mortality, and many other chronic diseases. Few studies have tackled this three-way interaction, mostly like because of its daunting complexity. Even after two decades of *APOE* research, the reasons behind the sex heterogeneity of *ApoE4* effects remains unclear. We hope our analysis will encourage further study of GxE for *ApoE* and sex.

15) About Abeta, only the ApoE3 mice responded. Isn't that surprising?

These results were unexpected, but lead us to seek a larger pattern of an *APOE4* ceiling effect. As noted above, this pattern was observed in other studies including O3 exposed *APOE*-TR animals.

16) Subsection “Air pollution nPM collection and exposure” need more detail about the chemical engineering used to prepare the samples. And "5 h/day, 3 days/week" for 8 weeks does not sound like "chronic exposure" to me.

As suggested, additional information about the chemical characterization of nPM was added to the paper. Chemical data were also added to the supplement.

17) Likewise, Results paragraph three: "Both analyses (DE, WGCNA) showed more nPM-responding gene responses for ApoE3 than E4." Isn't that surprising?

While surprising, our data shows this consistently.

18) In light of the senior author's long experience in genetics and molecular biology, it might be useful to use part of the Discussion to offer comments on the pathway analysis programs he uses: discussion of their epistomilogic status and the verity of the statistical analyses applied would head off charges of over-interpretation of the data presented here.

Additional discussion was added about the interconnectivity of the enriched pathway in paragraph ten of the Discussion section.

19) There seem to be emerging new theories re the chemical status of nanoparticle/microcavity relations. Can the authors use this in their Discussion?

Additional discussion was added about confounding effects of chemical and physical characteristics of PM on air pollution toxicity in paragraph twelve.